

# Comparison of HiL Control Methods for Wind Turbine System Test Benches

Lennard Kaven[1,2], Christian Leisten[1,2], Maximilian Basler[1,2], Moritz Schlösser[1], Uwe Jassmann[1,2], and Dirk Abel[1,2]

[1]Institute of Automatic Control, RWTH Aachen University, Campus-Boulevard 30, 52074 Aachen, Germany
[2]Center for Wind Power Drives, RWTH Aachen University, Campus-Boulevard 61, 52074 Aachen, Germany

**Correspondence:** Lennard Kaven (l.kaven@irt.rwth-aachen.de)

**Abstract.** The current test process in design and certification of wind turbines (WTs) is time and cost intensive, as it depends on the wind conditions and requires the setup of the WT in the field. Efforts are made to transfer the test process to a system test bench (STB) whereby an easier installation is enabled and the load can be arbitrarily applied. However, on a STB the WT is installed without rotor and tower and the remaining drive train behaviour acts differently to the WT drive train in the field.

The original behaviour must be restored by incorporating a Hardware-in.the-Loop (HiL) simulation into the operation of the STB. The HiL simulation consists of the virtual rotor and wind and the control of the applied loads. Furthermore, sensors as the wind vane and actors as the pitch drives, which are not present at the STB, are substituted by simulation models.

This contribution investigates suitable HiL control methods of the applied torque. Herein, we survey three methods of different complexity and compare them in terms of performance, actuator requirements and robustness. The simplest method

emulates the divergent inertia by classical control. A more complex method based on a reference model also considers the alternated dynamic behaviour of the drive train. Model predictive control (MPC) currently constitutes the most complex HiL method, as the MPC also includes future predictions of the driving torque behaviour. Our comparison identifies that increased complexity of the control method ensures enhanced preformance. WT drive train dynamics can be reproduced up to 1, 6, and 10 Hz for IE, MRC, and MPC, respectively. Yet, for higher control complexity, the requirements for the dynamic torque

proliferate and the controllers robustness to model deviations decreases.

## 1 Introduction

Wind energy represents the primary source of renewable energy. Although wind turbines (WTs) have been extensively studied throughout the last decades, the development of WTs of increasing size as well as the need to build wind farms in areas with

poor wind conditions along with the reigning cost pressure require further investigations. For this purpose, full-scale tests of the WTs on a system test bench (STB) enable detailed insights into the mechanical and electrical interaction of the WT system



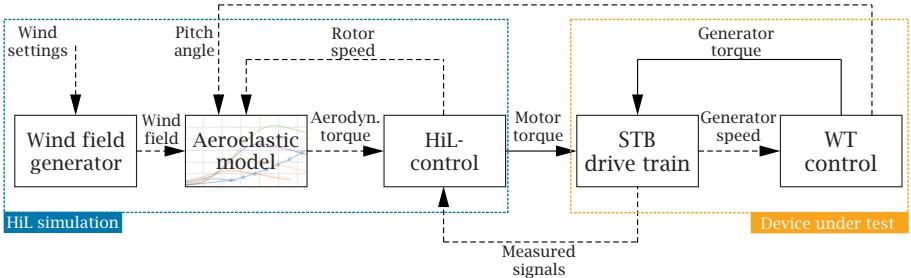

**Figure 1.** HiL system architecture based on Jassmann (2018).

components. Additionally, STBs have the potential to significantly change the certification process of WTs as on a STB with grid emulator, certification measurements can be carried out with arbitrary and hence reproducible wind and grid conditions.

A variety of test facilities possess a STB capable of performing full-scale tests for multi-MW WTs (e.g. Lindo Offshore Re-
newables Center[1], Clemson University[2] or Center for Wind Power Drives of RWTH Aachen University[3]). For performing test measurements at a STB, the WT is installed without rotor blades and tower, but with original WT controller. This configuration is refered to as "device under test" (DUT). During test executions, the WT controller sets the generator torque and the pitch angle of the rotorblades as in the field. To operate the DUT under realistic conditions, an aerodynamic torque is applied, which is calculated from a turbulent wind field and a rotor model of the WT. From a control perspective, the torque load application
is the most challenging part, therefore, bending moments and forces are not considered in this context.

In the field, the drive train of a WT oscillates in its eigenfrequencies due to the flexible rotor and turbulent inflow. A 3-mass oscillator was shown to sufficiently model the first two rotational eigenfrequencies and thus the drive train of a WT (Jassmann et al. (2015)). Similarly, modeling the drive train of the DUT on a STB is enabled by a 3-mass oscillator. In the DUT setup, the WT is mounted without its rotor and thus the drive trains inertia and the stiffness parameters differ to the WT in the field,
hence, impeding realistic tests. Only with a Hardware-in-the-Loop (HiL) control method that reproduces the WTs inertia and eigenfrequencies the potential of STBs can be fully exploited.

The operating mode of a HiL control is shown in Fig. 1. Depending on the test scenario, wind settings are defined and used by a wind field generator to create the turbulent inflow on an aeroelastic model of the rotor. Other inputs into this aeroelastic model include the pitch commands from the DUT controller as well as the rotational speed of the DUT. The HiL controller
uses the calculated aerodynamic torque to reproduce the WT behaviour as in the field. Depending on the applied HiL control method, feedback variables from the DUT to the HiL control differ. As illustrated by the generator torque and pitch angle feedback, the WT controller is active throughout HiL operation.

A variety of HiL methods have been proposed in the past. The inertia emulation (IE) method, initially presented by Jassmann et al. (2014), is suitable for emulating the missing inertia on a STB and is advantageous in terms of simplicity and commission-

---

[1]https://www.lorc.dk

[2]https://clemsonenergy.com

[3]https://www.cwd.rwth-aachen.de

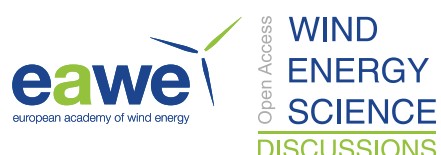

ing. Extensions to the IE method allow eigenfrequency emulation and are based on pole placement for state feedback control (Jassmann et al. (2015)) or a WT reference model (Neshati et al. (2016b), Jassmann (2018)). The previously listed control methods use classic PID or state feedback to control the rotational speed of the STB. More advanced control methods apply model predictive control (MPC) (Neshati et al. (2016a), Leisten et al. (2017)) or H-∞ control (Fischer et al. (2016), Neshati et al. (2017), Basler et al. (2020)).

Past contributions highlight the particular HiL method, but a comprehensive comparison of different HiL control methods has not yet been published. In this contribution, we oppose three HiL methods regarding their performance, the actuator requirements, and the robustness towards parameter uncertainties and dead time. The remainder of this paper is organized as follows: In section 2, three HiL methods are selected and described. In section 3, the selected HiL methods are compared based on analytical results and simulation results. Section 4 summarizes the advantages and restrictions of each method, interprets

the benefits for certain test scenarios, and outlines future work and potential for HiL control methods.

## 2 HiL control methods

In the following section, three HiL control methods with increasing complexity are briefly presented. The inertia emulation (Jassmann et al. (2014)) is the simplest method and serves as a benchmark. Second, as a method with increased complexity, the model reference control (MRC) with inertia and eigenfrequency emulation is chosen. This method is based on the contribution

by Jassmann (2018) and modified for speed control as suggested by Basler et al. (2020). From the advanced control methods, the MPC introduced by Leisten et al. (2017) is chosen for comparison. Although the MPC is not yet validated on a STB, advantages and drawbacks are outlined. For all three methods, we adopt the originally presented control approaches and implement only minor modifications.

Throughout the comparison of different HiL control approaches, the same HiL system architecture shown in Fig. 1 is used.

The aerodynamic excitation and the WT controller are neglected in this introduction of different HiL methods, but taken into account for the comparison in section 3. As a reference, we use the 3-mass oscillator model of a WT drive train, and for HiL controller synthesis the 3-mass oscillator model of a STB drive train parameterized as by Leisten et al. (2019b). A damping controller is applied to damp the STB eigenfrequencies and the damped STB drive train is referred to as "$G_{TB/wDmp}$".

### 2.1 Inertia emulation method

The most basic HiL method presented here is the IE (Jassmann et al. (2014)), of which a block diagram is shown in Fig 2. Here, a virtual rotor $I_{rot}$ representing the inertia difference between WT and STB is coupled to the STB drive train. The aerodynamic torque $T_{aero}$ accelerates the virtual rotor and the rotational speed $\omega_{ref}$ is used as reference for a speed controller. In contrast to Jassmann et al. (2014), instead of a P controller we implemented a PI controller, which is beneficial for higher performance and avoids a steady state error (a difference in rotational speed between virtual rotor and STB). The open loop and closed loop

transfer functions are shown in Fig. 3. The PI controller was tuned according to the Nyquist criterion with gain margin of 7 dB to compensate parameter deviations.



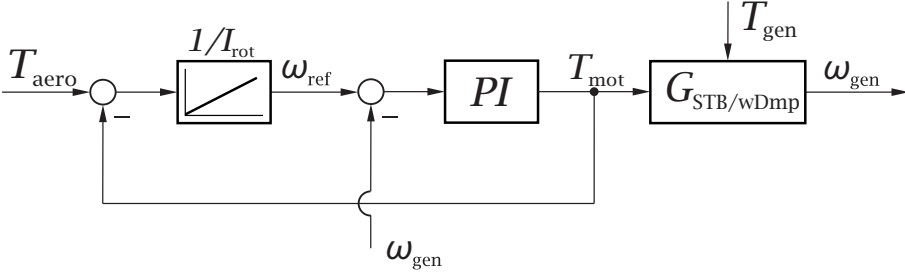

**Figure 2.** Block diagram of inertia emulation HiL control based on Jassmann et al. (2014).

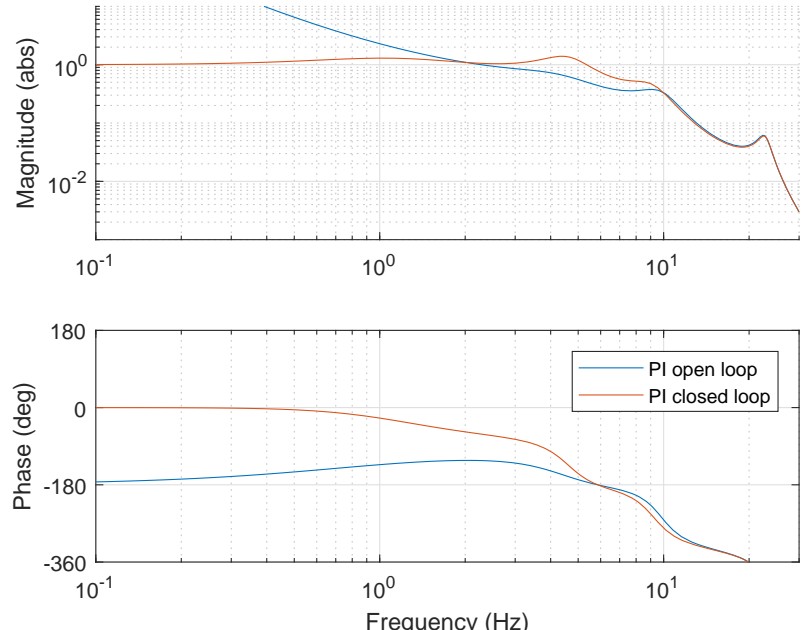

**Figure 3.** Bode diagram of transfer function from $\omega_{\mathrm{ref}}$ to $\omega_{\mathrm{gen}}$, used for controller synthesis of PI speed control.

### 2.2    Model reference control method with inertia eigenfrequency emulation

The MRC uses a model of the original WT's drive train to emulate the eigenfrequencies. A block diagram of the MRC is shown in Fig. 4. The aerodynamic torque $T_{\mathrm{aero}}$ and the generator torque $T_{\mathrm{gen}}$ are used as inputs and the virtual generator speed as reference for the speed controller. The very same PI controller as for the IE method can be used, since only the reference model is changed. Note that the generator torque signal is used as input and thus an additional interface with the DUT needs to be realized. However, the reference trajectory is more dynamic and includes the desired WT eigenfrequencies.



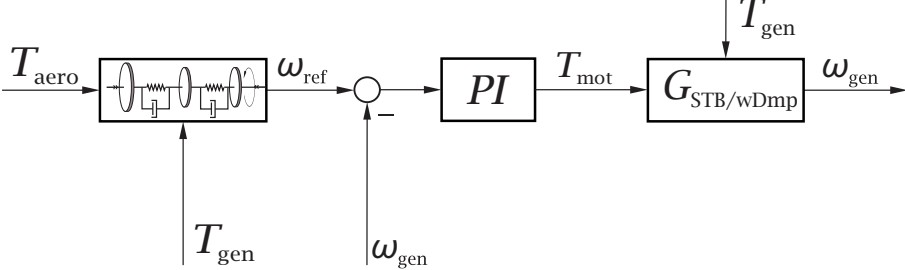

**Figure 4.** Block diagram of model reference HiL control based on Jassmann (2018) and Basler et al. (2020).

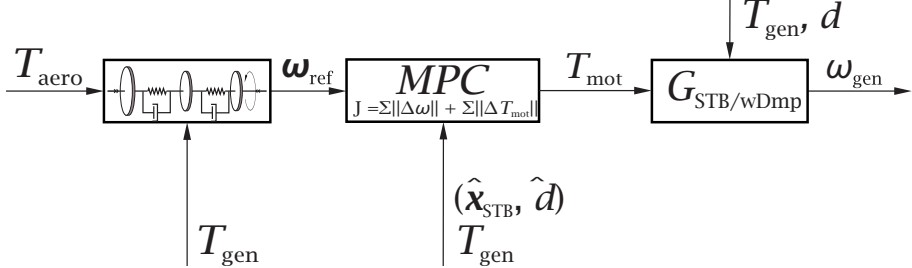

**Figure 5.** Block diagram of model predictive HiL control, based on Leisten et al. (2017). $\omega_{\mathrm{ref}}$ and $\hat{\mathbf{x}}_{\mathrm{STB}}$ represent vectors.

### 2.3 Model predictive control method

In this section, we illustrate the model predictive HiL control method introduced by Leisten et al. (2017), where the MPC design
is based on Maciejowski (2000). In MPC, a plant model and thus future knowledge is used to calculate optimal controller
actions. As a plant model we use the 3-mass oscillator of the STB introduced in section 1. Based on this model, the future
controller actions can be optimized with respect to a cost function J that penalizes predicted deviation of the controlled variable
$\omega_{\mathrm{gen}}$ and a change in the actuating variable $\Delta T_{\mathrm{mot}}$.

$$J = \sum_{i=1}^{N_p} ||\omega_{\mathrm{gen}}(k+i|k) - \omega_{\mathrm{ref}}(k+1|k)||^2 + \lambda \cdot \sum_{i=0}^{N_u-1} ||\Delta T_{\mathrm{mot}}(k+i|k)||^2 \tag{1}$$

Here, $\lambda$ is a tuning factor to weight the ratio between prediction deviation and controller action, while $N_p$ and $N_u$ represent
the prediction and control horizon, respectively. At each time step, the MPC minimizes the cost function subject to constraints
and sets the new optimal control value, the setpoint for the STB motor torque. The structure of the MPC is shown in the block
diagram in Fig. 5. Basically, the MPC replaces the PI reference controller in the previously introduced HiL methods. Note that
full state feedback is necessary for prediction, which requires state estimation. For disturbance rejection, the estimated state
vector $\hat{\mathbf{x}}_{\mathrm{STB}}$ is augmented with an estimate of the disturbance $\hat{d}$. A reference trajectory is calculated with a model of the WT's
drive train. Since no future knowledge for the inputs $T_{\mathrm{aero}}$ and $T_{\mathrm{gen}}$ is available, they are approximated by constant values.

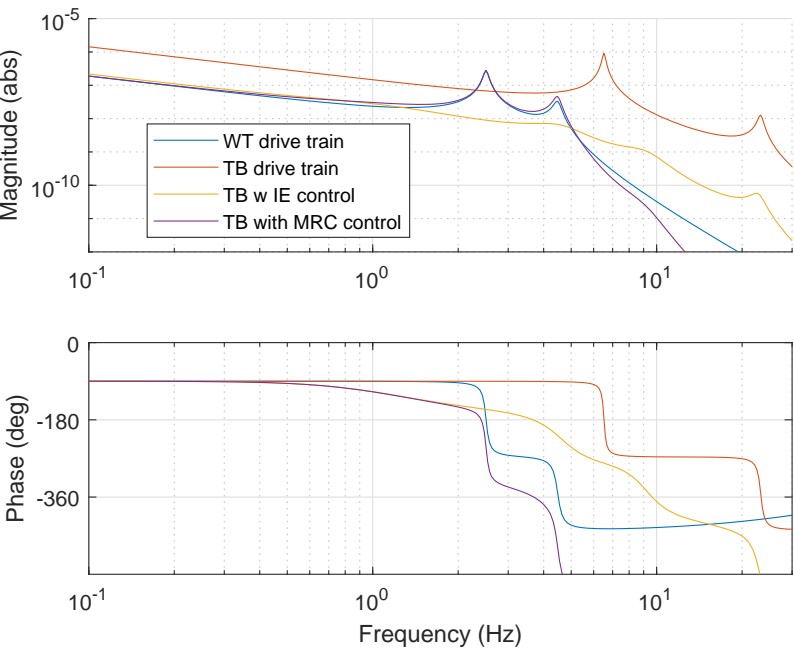

**Figure 6.** Bode diagram to compare performance of IE and MRC HiL control methods: The transfer function from input torque $T_{aero}$ / $T_{mot}$ to generator speed $\omega_{gen}$ is shown for wind turbine (WT), system test bench (STB), and STB with HiL control methods IE and MRC.

## 3   Comparison of HiL control methods

Based on the controller synthesis from section 2, analytical results and a comparison in terms of performance will be presented first. However, analytical comparison is limited to linear behaviour and does not consider the actuator constraints which are a major challenge in full-scale testing. Therefore, after an introduction of the simulation model and test settings, a comparison in terms of acuator requirements will be presented where also robustness of the HiL control methods is taken into account.

### 3.1   Analytical results

For the IE method and the MRC method, analytical results are presented. The transfer functions from input torque to generator speed is shown for the WT and STB drive trains in the bode diagram in Fig. 6. The eigenfrequencies of WT and STB are visible by peaks in the gain response. The transfer function for HiL controlled STB is shown for IE and MRC. The IE control achieves good accordance with the WT transfer function up to 1 Hz while the MRC control matches the desired behaviour up to 6 Hz. Thus, a reproduction of eigenfrequencies seems not achievable with IE control method.

Apart from eigenfrequency emulation, the option of controller validation is relevant. During the test procedure, the WT controller sets the generator torque. To receive realistic WT controller actions, a behaviour of the drive train as in the field is needed. To compare the effect of HiL methods on the drive train behaviour from the generator perspective, the open loop

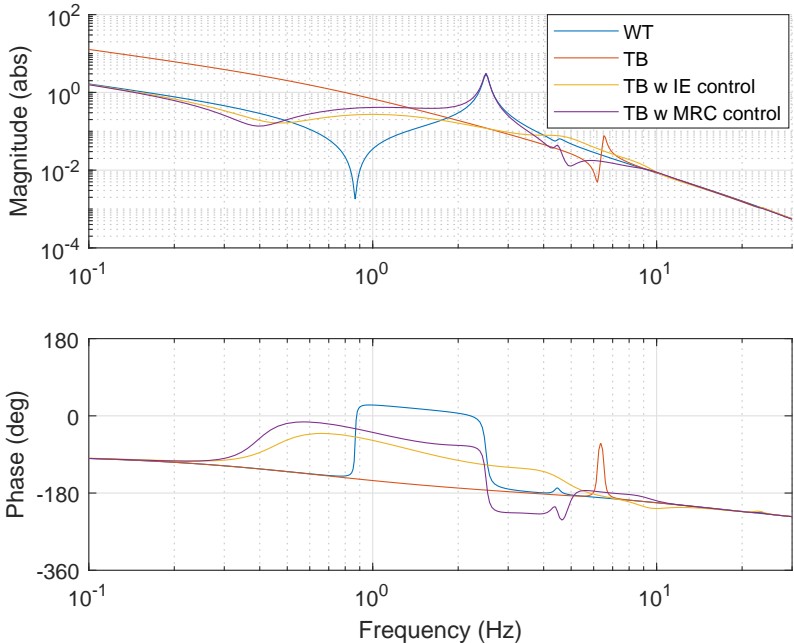

**Figure 7.** Bode diagram to compare performance of IE and MRC HiL control methods from generator perspective: The transfer function from generator torque $T_{\mathrm{gen}}$ to generator speed $\omega_{\mathrm{gen}}$ is shown for WT, STB and STB with HiL control methods IE and MRC.

transfer function of drive train and WT controller including a first order lowpass is shown in Fig. 7. To enable linear analysis, the nonlinear speed-torque diagram generally used in a WT control (Bossanyi (2003)) is approximated by a proportional gain at the section with the highest gradient. Sufficient accordance is achieved at low frequencies for both methods. The first WT eigenfrequency is visible at 2.5 Hz and is reproduced by the MRC method, whereas the anti-resonance at approximately 0.9 Hz

can not be reproduced by any of the two methods. Interestingly, at 2.5 Hz the phase response for STB with MRC method falls below 180 deg and thus causes instability for the WT controller feedback as the gain is still above 1. Instability can be overcome by using a higher order lowpass filter for the speed signal than the PT1 that was previously implemented for the WT controller. Nevertheless, for WT controllers actuating at this frequency, e.g., to damp the WT eigenfrequency, the deviation in phase response states a problem and complete controller validation seems not yet feasible with the MRC.


## 3.2 Simulation results

To compare the functionality of the HiL control methods, we simulate a production cycle with turbulent wind conditions at 20 m/s in MATLAB™Simulink. Apart from a STB drive train model and the HiL control method, the simulation includes an aerodynamic model of the rotor (Marnett et al. (2014)) and a WT controller model (Leisten et al. (2019a)). By substituting





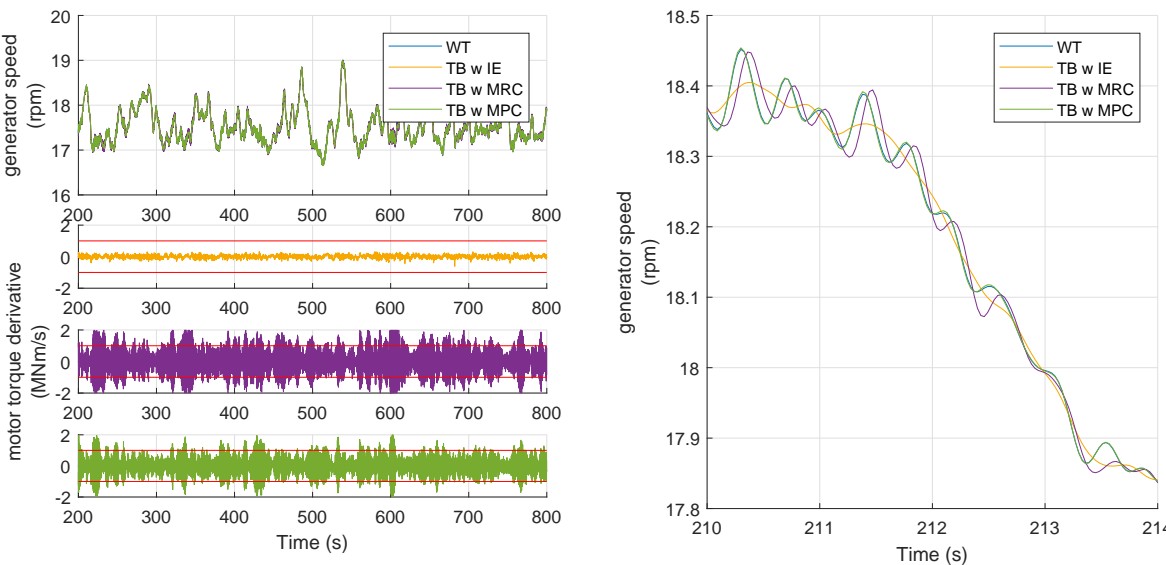

**Figure 8.** Simulation results at 20 m/s turbulent inflow. Left: High wind speeds lead to exceeding of the STB derivative torque limit for MRC and MPC methods. Right: different performance of HiL methods in the time domain.

the STB drive train and the HiL control method by the WT drive train, the same simulation can be used to generate the WT behaviour as a reference.

HiL operation is possible for all three control methods. In the time domain, differences to the baseline results of the WT simulation can be observed in Fig. 8. The IE method is capable of reproducing the WT generator speed roughly, but fails to follow the oscillations in the eigenfrequencies. With MRC, the amplitudes match in higher frequencies, but perfect conformity
is prevented by a phase shift. The MPC shows the best match in time domain, as the amplitudes match even better than the MRC and no phase shift is present. A limitation for performance on STB is the motor torque derivative. With IE control, the actuator requirements are below 0.5 MNm/s but for MRC and MPC control, the extrema rise above 2 MNm/s. The comparison results concerning performance and actuator requirements are summarized in Table 1. Analysis of the simulation results in the frequency domain is undertaken using the power spectral density estimate (PSD) of the generator speed. Fig. 9 shows the PSD
for each HiL method in comparison to the WT. The PSD is used instead of Fourier transform to include oscillations at high frequencies, where the amplitudes are habitually lower. With the PSD, we can examine to what extend the WT's dynamics are reproduced on the STB in HiL operation. The PSD of the WT simulation results shows that oscillations occur in the WT drive train's eigenfrequencies at 2.5 and 4.5 Hz, but also at lower frequencies. The lower frequency share derives from excitation through the tower shadow (3P frequency). For IE and MRC methods, the simulation results match the findings from analytical
comparison, although some accordances in the frequency domain can be observed for the IE method as well. For the MPC, eigenfrequency emulation is achieved up to 10 Hz.


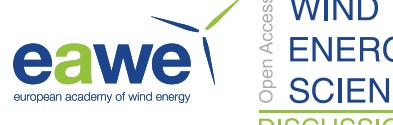

**Figure 9.** Power spectral density estimate of the generator speed for simulation results with the different HiL methods. Top left: PSD of WT and STB with IE method; Top right: PSD of WT and STB with MRC method; Bottom: PSD of WT and STB with MPC method.

In the following, controller performance with active constraints of +/-1 MNm/s motor torque derivative will be compared. Additionally, to compare the robustness against parameter deviations and time delay, we implemented a perturbed STB drive





**Table 1.** Comparison of IE, MRC and MPC methods in terms of perfomance, actuator requirements and robustness.

| Method | Performance | Actuator requirements | Robustness |
|---|---|---|---|
| IE | 1 Hz | <0.5 MNm/s | robust against parameter deviation and dead time |
| MRC | 6 Hz | >2 MNm/s iterative controller derating process | robust against parameter deviation and dead time |
| MPC | 10 Hz | >2 MNm, limits can be included in constraints | sensitive to parameter deviation |

train with decreased stiffness coefficients $c_i$* as stated in (2) and a time delay of 10 ms on the motor torque input $t_{mot}$.

$$
c_1^* = c_1 \cdot 0.8
$$
$$
c_2^* = c_2 \cdot 0.8
$$


(2)

With these simulation parameters, HiL operation fails for the MRC and MPC method as exceeding the motor torque derivative contraints. For MPC, the actuator limits can be included as optimisation constraints. For MRC, derating the PI speed controller is required. Due to the saturation, analysis with the Nyquist criterion is not available and an iterative procedure is forced. For the following comparison, simulation results are shown with 60% derated MRC speed controller. The results for simulation with

perturbations and active contraints are shown in Fig. 10. Again, we use the PSD to compare the HiL controller performance. For the IE method, the accordance of PSD for WT and HiL simulation results has slightly improved. The exitation of frequencies at 3.5 and 4.5 Hz might be a result of the now ill-tuned subsidiary damper. The performances of the MRC and MPC methods have decreased. Although WT eigenfrequencies still match, excitation of frequencies at 7 Hz occurs. For the MPC method, the accordance of WT eigenfrequencies has declined due to the model perturbations. These findings complete the comparison

results summarized in Table 1. Whereas the IE method is advantageous in terms of robustness and actuator requirements, it lacks performance. Similarly, the MRC is beneficial when it comes to robustness, while higher performance is achieved at the expense of higher actuator requirements. Finally, the MPC method is favourable concerning the performance and handling of actuator constraints, but is sensitive to parameter deviations. However, as no method is outpacing the others in all criterions, the preferable HiL control method must be chosen according to the particular usecase. Here, the undertaken comparison helps

to find the appropriate method.

## 4 Conclusions

In this contribution, we compared HiL control methods for operating a WT STB. The HiL control methods chosen for comparison included the IE method, the MRC method and the MPC method. We illustrated that for the IE and MRC methods, the design process is simple, but the MRC method needs iterative derating, when actuator constraints are taken into account.

Further, we ponited out that the design process of the MPC method is more complex, but constraints can be considered in the optimization. For the MRC and the MPC methods, advantages in the performance can be observed in the frequency domain,



**Figure 10.** Power spectral density estimate (PSD) of the generator speed for simulation results with three HiL methods and model perturbations and active constraints. Top left: PSD of WT and STB with IE method; Top right: PSD of WT and STB with MRC method; Bottom: PSD of WT and STB with MPC method.

which we derived analytically and based on simulation results. For unconstrained HiL simulation of a production cycle, the actuator requirements of IE method is relatively low compared to the MRC and MPC methods. A drawback of the MPC method remains, as the perfomance behaves very sensitively to parameter deviations.



Our comparison proves that although IE and MRC methods might be satisfactory for some tests in HiL operation, MPC offers advantages in handling of constraints and might increase performance of HiL tests, which is especially relevant for WTs with higher eigenfrequencies. Additionally, we evidenced that IE and MRC methods are not sufficiently accurate to enable complete WT controller validation on STBs. However, the MPC method constitutes a promising alternative for validation and hence should be focus of further research. Yet, the MPC implementation must include the design of a state estimator that deals

with parameter uncertainties to tackle the sensitivity of the MPC to model deviations. Altogether, the analysis of HiL methods proffers a guideline for future STB control approaches, thus paving the way for WT certification off the field.

*Author contributions.* CL conceived the idea to examine a detailed comparison. CL, MB and UJ developed the HiL methods and models of the HiL methods considered for domparison. LK contributed the simulation results and conducted the comparison in discussion with all authors. LK prepared the manuscript and MS assisted in the manuscript preparation. DA supervised the findings of this work.

*Competing interests.* The authors declare that no competing interests are present.

*Acknowledgements.* The depicted research is part of the project CertBench – 'CertBench - Systematic validation of system test benches based on type testing of wind turbines' and funded by the German Federal Ministry of Economic Affairs and Energy (0324200A).

Supported by:

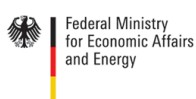

Federal Ministry
for Economic Affairs
and Energy

on the basis of a decision
by the German Bundestag



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
