# Peer review of "Comparison of HiL Control Methods for Wind Turbine System Test Benches"

_Wind Energy Science, 2020_

## Referee Comment (RC1) · Anonymous Referee #1 · 13 Apr 2020

The article proposes to assess different methods in the implementation of HIL for test benches. Very little information is provided on the drivetrain being tested or the wind turbine that is modeled. Also there are limited details provided on the implementation of each of the controllers. So the results presented in this article are not reproduceable. Further an evaluation of three standard control approaches on an unspecified system or plant is not novel and it is not justified for publication in its present state. Specific areas of the article that needs improvement are the following:

1) The abstract needs to be re-written to clearly articulate what novelty has been investigated and what was found. 2) Page 2, line 35: "Only with a Hardware-in-the-Loop (HiL) control method that reproduces the WTs inertia.....". Please define what aspects of the control system do you define as HiL and whether it operates on a realtime platform? 3) The introduction seems to heavily rely on prior publications by one or more of the authors of this article. The references should be broadened to include other relevant publications in the field. 4) Page 3, line 71: How do you quantify the inertia difference between WT and STB? Do you need to have the aeroelastic model of the turbine from the manufacturer? 5) Do you only model torsional inertia of the drivetrain or is bending inertia also considered? 6) Page 4: Line 80: What are the desired WT eigenfrequencies that you seek to represent in the MRC model and where are they shown? 7) Page 5: What is the sampling frequency and time step for equation 1 ? 8) Page 5: How do you determine the Lambda parameter and how do you ensure convergence to the required rotor speed? 9) The Bode plots (fig 6 and 7) do not seem to capture the WT behavior, but perhaps this is because of the specifics of the implementation of the methods? 10) Page 7, line 122: "turbulent wind conditions at 20 m/s...". The methods outlined mention generator torque control. It is not clear how the effects of pitch control on the rotor aerodynamics was accounted for? 11) It is unclear what Figure 8 is comparing. The legends on the left do not match the plots. 12) Figures 9: It is not clear why generator speed PSD is focused on as the only result for validating WT dynamics. At 20 m/s mean wind speed, the generator speed should not show high variations so long as the wind speed stays above rated? Why are there no plots shown at 7m/s or 9m/s etc where there will be a variation in generator speed based on the torque control? Also what about other system responses such as shaft loads? 13) Stated conclusions such as the "excitation of frequencies may be a result of the ill-tuned damper" and " MPC method constitutes a promising alternative for validation" seem to be very non-specific and hardly convincing.

---

## Referee Comment (RC2) · Amir R. Nejad (Referee) · 13 Apr 2020

This article presents the application of HiL for drivetrain testing in onshore wind turbines. The article is well-structed and topic is of interest. Here are my suggestions to improve the paper quality: -Title: please add "drivetrain" to the title. -Apart from the inertia and stiffness which are considered by the 3-mass model, there are "external excitations" in particular "tower shadow" or "blade passing frequency" (often shown as 1P, 3P, 6P,...). How authors have modelled them? I would expect to see some peaks less than 1 Hz for 3P (with rated speed of 17.5 rpm, the 3P would be 0.85 Hz) which could be seen in Fig. 6 if it is modelled. -Abstract: some of the abbreviation are not defined (IE, MRC) in the abstract. -Fig 1: please clarify what are "measured signals" in this figure. -Introduction: authors can extend the introduction by referring to several

works on the HiL application in wind industry, for example for offshore turbines. HiL has been used in testing the offshore wind turbines. Please also extend the literature review for the control methods used in this paper – specially refer to their applications in other fields. -Page 3 (65) "As a reference, we use the 3-mass oscillator model of a WT drive train, and for HiL controller synthesis the 3-mass oscillator model of a STB drive train parameterized as by Leisten et al. (2019b).Âż Please mention the main futures of the WT and drivetrain used as reference case in this study. -Fig 6: please discuss the results more in details, specially why the results for each method differs to others. - Please consider including a nomenclature listing all abbreviations.

---

## Author Comment (AC1) · 25 May 2020

We thank the anonymous referee #1 for the extensive review. We considered the remarks very helpful to further improve the quality of the manuscript.

General remarks:

Anonymous referee general remarks: Very little information is provided on the drive-train being tested or the wind turbine that is modeled. Also there are limited details provided on the implementation of each of the controllers. Further an evaluation of three standard control approaches on an unspecified system or plant is not novel and it is not justified for publication in its present state.

Author's response to general remarks: The plant is a multi-MW experimental system

test bench for wind turbines. The choice of rather fundamental control approaches is justified, since the HiL methods aim at experimental validation, impeded by the dimensions of the test bench and safety concerns. To keep the focus on the HiL control methods, we omitted most of the details on the device under test and the original wind turbine. The authors agree with the referee that further information is beneficial for interpretation and reproduction of the results and the corresponding details have been added to the text. Finally, a comparison of control approaches, in this case three, for a multi-MW system test bench with HiL functionality has not been published. We hope that with the following changes, the relevance of the findings for an international audience becomes clearer.

Further remarks:

Anonymous referee remark (1): The abstract needs to be re-written to clearly articulate what novelty has been investigated and what was found.

Author's response on remark (1): The quantitative comparison of the HiL control methods is novel as it offers a guideline for selecting the appropriate method for respective testing demands. The different performances of the HiL control methods were already included in the abstract of the discussion paper. Changes in the manuscript: An explicit statement of the novelty has been added to the abstract (l.15): "Our quantitative comparison of HiL control methods facilitates selecting the appropriate method for respective testing and certification demands."

Anonymous referee remark (2): Page 2, line 35: "Only with a Hardware-in-the-Loop (HiL) control method that reproduces the WTs inertia.....". Please define what aspects of the control system do you define as HiL and whether it operates on a realtime platform?

Author's response to remark (2): In general, HiL systems substitute missing components by simulation models and connect the simulation models to the actually present components. In the present application, the actually present components, or "hardware", is not the control system, but the device under test, as depicted in Fig. 1. The presented HiL system consists of the wind simulation, the aerodynamic simulation, and the HiL control, as depicted in Fig. 1. The simulation models and the HiL control are executed on a real-time platform DS1006. Changes in the manuscript: To clarify the HiL system setup, we took various measures: (a) A definition of HiL systems in general has been added to the text (l.37): "HiL systems substitute missing components by simulation models and connect the simulation models to the actually present components."; (b) An explicit determination of the substituted wind turbine's components has been added (l.37):"On a full-scale WT STB, the HiL system needs to replace the wind inflow and the aerodynamic properties of the rotor, and restore the mechanical properties of the original WT drive train."; (c) In Fig. 1, we changed the caption for the aeroelastic simulation and the HiL control from "HiL simulation" to "HiL software". The demand for real time capability of the aerodynamic model has been added to the text (l.38): "The aerodynamic model needs to be executable in real-time, because the aerodynamic torque is calculated online during test procedure."

Anonymous referee remark (3): The introduction seems to heavily rely on prior publications by one or more of the authors of this article. The references should be broadened to include other relevant publications in the field.

Author's response to remark (3): Only a limited number of publications for HiL control methods on full-scale test benches exist of which the relevant publications had been summarized in the introduction of the discussion paper. The literature overview has been extended to HiL control for scaled system test benches, comparison of control methods without HiL functionality, and HiL control for different test benches. (a) HiL control for system test benches: Song et al. (2005): "Emulation of output characteristics of rotor blades using a hardware-in-loop wind turbine simulator"; Schkoda et al. (2015): "A Hardware-in-the-Loop Strategy for Control of a Wind Turbine Test Bench"; (b) comparison of control methods without HiL functionality: Thomsen et al. (2011): "PI control, PI-based state space control, and model-based predictive control for drive

systems with elastically coupled loads - A comparative study"; (c) HiL control for different test benches: Thun (1984): "Verfahren zum Simulieren von Trägheitsmomenten und geregelter Prüfstand zur Durchführung des Verfahrens [Procedure for simulating moments of inertia and a controlled test bench for validation of the procedure]"; Bayati et al. (2018): "A wind tunnel/HIL setup for integrated tests of Floating Offshore Wind Turbines"

Anonymous referee remark (4): Page 3, line 71: How do you quantify the inertia difference between WT and STB? Do you need to have the aeroelastic model of the turbine from the manufacturer?

Author's response to remark (4): The calculation of the inertia difference is based on the 3-mass oscillators. The inertia difference takes into account the entire rotational inertia of the wind turbine's and the tests benches drive trains. The 3-mass oscillators are derived from a multi-body simulation model according to Baseer et al. (2019). However, an aeroelastic model of the wind turbine from the manufacturer is required for HiL operation. It might directly be used for HiL simulation, but real-time execution must be established for the HiL models. At the CWD HiL sytem, the aeroelastic properties of the rotor blades are instead transferred to the real-time capable aeroelastic simulation model RAISE. Changes in the manuscript: An equation for calculating the inertia difference has been added to the text (l.71). Additional information on how the aerodynamic model and the models for the drive train are derived has been added to section 3: "The dominant eigenfrequencies and the drive train parameters have been derived from a multi-body simulation (Baseer et al., 2019). Drive train parameters of the 3-mass oscillators for WT and STB can be found in the appendix (Table A2). To model the aerodynamic properties, we use the rotor-aeroelastic-integrated-simulation-environment (RAISE) introduced by Marnett et al. (2014). RAISE takes into account elastic blades and tower shadow excitation and is capable of real time execution." Please note that Table A2 of the manuscript's appendix is attached to this document

Anonymous referee remark (5): Do you only model torsional inertia of the drivetrain or is bending inertia also considered?

Author's response to remark (5): Inside the aeroelastic rotor model also bending moments and forces are calculated, where a radially distributed mass of the rotor blades and thus bending inertia is considered. Application of non-torque loads is possible at the system test bench by means of a wind load unit with incorporated control. However, as stated in the text (ll.30,f), only application of torque loads on the drive train is considered in the HiL context. Changes in the manuscript: Details on the aeroelastic simulation are not focus of this contribution and thus the information on bending inertia is not added to the text.

Anonymous referee remark (6): Page 4: Line 80: What are the desired WT eigenfrequencies that you seek to represent in the MRC model and where are they shown?

Author's response to remark (6): The MRC method enables reproduction of the dominant rotational eigenfrequencies of a wind turbine drive train. As stated in ll.31,f. it is regarded as sufficient to reproduce the two dominant eigenfrequencies, which is why a 3-mass oscillator is used. For the wind turbine model used throughout this contribution, the dominant eigenfrequencies are the first edgewise blade mode and the first drive train mode. Changes in the manuscript: Information on the modeled wind turbine's eigenfrequencies has been added to the text in section 3: "The two dominant eigenfrequencies of the WT drive train are the first edgewise blade mode and the first drive train torsional mode, located at 2.5 and 4.6 Hz, respectively." A parameter table for details on the wind turbine and system test bench drive trains has been added to the appendix. (attached)

Anonymous referee remark (7): Page 5: What is the sampling frequency and time step for equation 1 ?

Author's response to remark (7): Sampling time and step size of the MPC are 1ms and 10ms, respectively. Changes in the manuscript: Information on step size has been

added to a table for controller parameters in the appendix (attached). Information on the sampling time has been added to the text (l.91): "After each time step of 1ms, the MPC minimizes (. . .)"

Anonymous referee remark (8): Page 5: How do you determine the Lambda parameter and how do you ensure convergence to the required rotor speed?

Author's response to remark (8): Lambda is tuned using the undisturbed simulation model and a synthetic load case. Convergence is ensured by the state augmentation with the disturbance estimate, as mentioned in the text (l.94): "For disturbance rejection. . ."). Changes in the manuscript: Tuning procedure for lambda is elucidated as follows: "To determine the optimal tuning factor lambda we conduct simulations for various tuning factors with a synthetic load case. The load case reproduces the startup procedure of a WT, e.g. a torque ramp up to 1 MNm for the aerodynamic torque T_aero followed by a step of 1 MNm for the generator torque T_gen. The simulation results are evaluated with respect to the integral for the squared error of the controlled variable omega_gen. The smallest value of the integral reveals the optimal tuning factor lambda." The convergence by state augmentation has been clarified in the text (l.94): "The estimated state vector is augmented with an estimate of the disturbance for disturbance rejection and to guarantuee convergence of the generator speed."

Anonymous referee remark (9): The Bode plots (fig 6 and 7) do not seem to capture the WT behavior, but perhaps this is because of the specifics of the implementation of the methods?

Author's response to remark (9): Indeed, the wind turbine's behavior is not exactly reproduced by either of the HiL control methods. This is expected for the IE method, because no information on the wind turbine's eigenfrequencies is provided. For the MRC method, only small deviations in the magnitude response prevent exact reproduction up to 6 Hz. Fig. 3 reveals that the small deviations are caused by the implemented speed controller. However, the bode plots show that MRC method matches the

wind turbine's behaviour more accurately, e.g. in terms of eigenfrequency emulation. Changes in the manuscript: The analysis of the analytical results (ll.103-107) has been complemented by a more detailed explanation of the results and an analysis, why exact reproduction of the wind turbine's behavior is not achieved: "Analytical results are presented as transfer functions. Fig. 6 depicts a bode plot for the transfer function from input torque to generator speed for the WT and STB drive trains. The inertia difference and eigenfrequencies of the WT and STB drive trains are observable by an offset towards low frequencies and peaks in the magnitude response, respectively. Additionally, the transfer function for the HiL controlled drive train of the STB is shown for IE and MRC methods. The transfer function with IE method achieves good accordance with the transfer function of the WT up to 1 Hz. The accordance is enabled by the virtual rotor that lowers the magnitude response of the STB until it matches the transfer behaviour of the WT drive train. For frequencies above 1 Hz, the transfer functions mismatch as the first WT drive train eigenfrequency dominates. Thus, a reproduction of the eigenfrequencies seems not achievable with IE method. In contrast to the IE method, the MRC method matches the desired behaviour in the magnitude response up to 6 Hz, despite small deviations. Fig. 3 reveals that the small deviations in magnitude response are caused by the implemented speed controller. The reference model of the WT drive train provides the required WT eigenfrequencies, at the same time eliminating the inertia difference. For frequencies above 6 Hz, the bandwidth of the speed controller becomes a limiting factor and transfer functions of the WT drive train and the STB drive train with MRC method drift apart. However, the reproduction of the WT eigenfrequencies seems possible with the MRC method. The phase response of none of the methods exactly matches the desired transfer function for frequencies above 0.5 Hz. The deviation in phase response is uncritical for HiL operation if WT pitch control provides sufficient phase margin."

Anonymous referee remark (10): Page 7, line 122: "turbulent wind conditions at 20 m/s...". The methods outlined mention generator torque control. It is not clear how the effects of pitch control on the rotor aerodynamics was accounted for?

Author's response to remark (10): Pitch control affects the aerodynamic simulation and thus the excitation of the drive train, as shown in Figure 1. Hence, the effects of pitch control is observable on the system test bench. The pitch actuator is substituted by a simulation model, e.g. a first order lag element. We regarded this information as uncritical, but added it for sake of completeness. Changes in the manuscript: Information on the pitch actuator model has been added to section 3.2: "For the pitch actuator model, a first order lag element is used and the actual pitch angle is passed to the WT controller and the aerodynamic simulation."

Anonymous referee remark (11): It is unclear what Figure 8 is comparing. The legends on the left do not match the plots.

Author's response to remark (11): As explained in the caption, Fig. 8 shows the performance of the HiL control methods in the time domain. Additionally, the motor torque derivative is plotted to point out the exceedance of constraints. We insist that the legend matches the plots, although the WT trajectory is covered by the trajectory for the system test bench with MPC method and is thus hardly visible. In addition, a typo (TB instead of STB) has been corrected. Changes in the manuscript: To facilitate the reading, Fig. 8 has been rearranged such that the performance is shown on the left and exceedance of the derivative torque constraints on the right. Additionally, the caption has been extended such that it explains the exceedance of the constraints more clearly: "(...) Derivative motor torque demand of the HiL control methods. High wind speeds lead to exceeding of the STB derivative torque limit for the MRC and MPC methods." The fact that wind turbine simulation results are covered has been added to the text (l.128): "... observed in Fig. 8. Note that the WT trajectory is hardly visible since covered by the trajectory for the STB with MPC method."

Anonymous referee remark (12): Figures 9: It is not clear why generator speed PSD is focused on as the only result for validating WT dynamics. At 20 m/s mean wind speed, the generator speed should not show high variations so long as the wind speed stays above rated? Why are there no plots shown at 7m/s or 9m/s etc where there will be

a variation in generator speed based on the torque control? Also what about other system responses such as shaft loads?

Author's response to remark (12): We focused on the generator speed for comparison, as it is the controlled variable. Of course, for wind speeds above rated, the generator speed does not show high deviations from the rated generator speed, but oscillations at high frequencies remain. When wind speeds fall below rated, variations in generator speed are indeed higher, but also slower. For this reason, simulation results are shown for an average wind speed of 20 m/s to achieve higher aerodynamic torques and thus eigenfrequency excitation of the generator speed. Consequently, the difference between the HiL methods is visible more clearly than at lower wind speeds, e.g. in part load. Simulations in part load have been conducted, but are not presented in the manuscript, as they do not reveal additional findings. The finding that dynamics match for low frequencies is included in Figures 9 and 10. All three HiL control methods are suitable for reproducing the generator speed dynamics at this bandwidth. Changes in the manuscript: The reason for presenting simulation results for 20m/s wind speed has been added to the text (l.126): "An average wind speed of 20 m/s is chosen to achieve higher aerodynamic torques and thus higher eigenfrequency excitation of the generator speed. Consequently, the difference between the HiL methods is visible more clearly than at lower wind speeds, e.g. in part load. All three HiL control methods are suitable for reproducing the slow generator speed variations, caused by alternating wind speeds in part load. Simulations in part load have been conducted, but are not presented in this contribution, since the finding that dynamics match for low frequencies is included in Figures 9 and 10."

Anonymous referee remark (13): Stated conclusions such as the "excitation of frequencies may be a result of the ill-tuned damper" and "MPC method constitutes a promising alternative for validation" seem to be very non-specific and hardly convincing.

Author's response to remark (13): The conclusion "excitation of frequencies may be a result of the ill-tuned damper" refers to the simulation results of the perturbed system

test bench with inertia emulation method. The minor increase in excitation of eigen-frequencies was observed shortly before manuscript submission. In the meantime, we further investigated the excitation of eigenfrequencies in the simulation results. We identified the PI speed controller being the reason for increased eigenfrequency excitation by calculating the closed loop transfer function of the PI speed controller for perturbed parameters of the system test benches drive train. Changes in the manuscript: We corrected the explanation for increased eigenfrequency excitation with perturbed parameters for inertia emulation method (ll.151,f):"The improved accordance is caused by an amplitude amplification of the speed controller. The amplitude amplification is visible in Fig. 3 for the original PI speed controller at 4.5 Hz. Due to the changed STB parameters, the closed loop performance of the speed controller shows a further increased amplitude amplification of frequencies at 3.5 and 4.5 Hz." The conclusion concerning the MPC method has been refined by adding further explanations in l.173,ff: "(. . .) we evidenced that IE and MRC methods are not sufficiently accurate to enable complete WT controller validation on STBs, especially due to phase shift. In contrast, the MPC method avoids phase shift and thus constitutes a promising alternative for HiL control."
* * *
**Table A1.** Controller parameters for IE, MRC and MPC methods

| HiL Control method | Controller parameters |
| --- | --- |
| IE method | $K_P = 10e6$; $K_I = 50e6$ |
| MRC method | $K_P = 10e6$; $K_I = 50e6$ |
| MPC method | $t_{step} = 10ms$; $N_U = 45$; $N_P = 13$; $\lambda_{MPC} = 1e\text{-}4$; $t_{sampling} = 1ms$ |

**Fig. 1.**

**Table A2.** 3-mass oscillator drive train parameters for WT and STB with DUT

| Drive train parameter | WT | STB with DUT | [unit] |
|---|---|---|---|
| $I_1$ | 3.42e6 | 8e4 | $[\mathrm{kgm}^2]$ |
| $I_2$ | 4.06e6 | 3.3e4 | $[\mathrm{kgm}^2]$ |
| $I_3$ | 1e6 | 1e6 | $[\mathrm{kgm}^2]$ |
| $c_{12}$ | 5.2e8 | 3.7e8 | $[\mathrm{Nm/rad}]$ |
| $d_{12}$ | 1.3e6 | 2e4 | $[\mathrm{Nm/(rad/s)}]$ |
| $c_{23}$ | 6.2e8 | 2.3e8 | $[\mathrm{Nm/rad}]$ |
| $d_{23}$ | 1.3e6 | 2e5 | $[\mathrm{Nm/(rad/s)}]$ |
| $f_{eig,1}$ | 2.5 | 6.5 | $[\mathrm{Hz}]$ |
| $f_{eig,2}$ | 4.6 | 23 | $[\mathrm{Hz}]$ |

**Fig. 2.**

---

## Author Comment (AC2) · 25 May 2020

We thank the referee Amir R. Nejad for the detailed review. We considered the remarks very helpful to further improve the manuscript's quality.

General remarks:

Referee general remarks: This article presents the application of HiL for drivetrain testing in onshore wind turbines.The article is well-structed and topic is of interest.

Author's response to general remarks: We thank the referee for the positive feedback.

Further remarks:

Referee remark (1): Title: please add "drivetrain" to the title.

[Figure]

Author's response on remark (1): Indeed, the HiL control methods focus on the drive train of the system test bench. We agree that adding the keywords to the title will increase the paper's impact. Changes in the manuscript: The title has been changed to "Comparison of Hardware-in-the-Loop Control Methods for Wind Turbine Drive Trains on System Test Benches"

Referee remark (2): Apart from the inertia and stiffness which are considered by the 3-mass model, there are "external excitations" in particular "tower shadow" or "blade passing frequency" (often shown as 1P, 3P, 6P,: : :). How authors have modelled them? I would expect to see some peaks less than 1 Hz for 3P (with rated speed of 17.5 rpm, the 3P would be 0.85 Hz) which could be seen in Fig. 6 if it is modelled.

Author's response to remark (2): Fig. 6 shows the transfer function of the wind turbine's drive train, i.e. aerodynamic torque to generator speed. Thus, external excitations are not visible here. However, external excitations as the tower shadow are included in the aerodynamic rotor model. As the topic of the paper are the control methods, we had left out most of the details concerning the aerodynamic simulation in the discussion paper. We now decided, that for sake of interpretation of the results and reproducibility, information on the aerodynamic rotor model is added to the text. Changes in the manuscript: We added the requirements for an aerodynamic rotor model to section 1 (l.38): ("The requirements for the aerodynamic model depend on the use case. For simple tests, a CP-lambda curve might be sufficient, but to realistically reproduce oscillations, an aeroelastic model that considers blade elasticity and tower shadow is required. Additionally, the aerodynamic model needs to be executable in real-time, because the aerodynamic torque is calculated online during test procedure."). Secondly, details on the aerodynamic model used for simulation results is added to section 3 (l.101): "To model the aerodynamic properties, we use the rotor-aeroelastic-integrated-simulation-environment (RAISE) introduced by Marnett et al. (2014). RAISE takes into account elastic blades and tower shadow excitation and is capable of real time execution."

Referee remark (3): Abstract: some of the abbreviation are not defined (IE, MRC) in the abstract.

Author's response to remark (3): This has been corrected. The abbreviations of the HiL control methods IE and MRC are now introduced in the abstract.

Referee remark (4): Fig 1: please clarify what are "measured signals" in this figure.

Author's response to remark (4): As stated in the text, the measured signals differ "depending on the applied HiL control method" (l.40,f). Deducting the measured signals from the respective HiL control method is possible, though not intuitive. Hence, we added the possible measured signals to Fig. 1 as suggested by the referee. Changes in the manuscript: In Fig. 1, we have changed the label of the feedback variables from DUT to HiL control from "Measured signals" to "Generator speed, (Generator torque)" to indicate that all of the HiL control methods use the generator speed, but only some of the HiL control methods use the generator torque. Additionally, this information has been clarified in the text.

Referee remark (5): Introduction: authors can extend the introduction by referring to several works on the HiL application in wind industry, for example for offshore turbines. HiL has been used in testing the offshore wind turbines. Please also extend the literature review for the control methods used in this paper – specially refer to their applications in other fields.

Author's response to remark (5): We thank the referee for pointing at the HiL application on a test bench for floating offshore wind turbines. We included this work in the literature overview. Furthermore, we extended the literature overview to HiL control for scaled system test benches, comparison of control methods without HiL functionality, and HiL control for different test benches. (a) HiL control for system test benches: Song et al. (2005): "Emulation of output characteristics of rotor blades using a hardware-in-loop wind turbine simulator"; Schkoda et al. (2015): "A Hardware-in-the-Loop Strategy for Control of a Wind Turbine Test Bench"; (b) comparison of control methods without

HiL functionality: Thomsen et al. (2011): "PI control, PI-based state space control, and model-based predictive control for drive systems with elastically coupled loads - A comparative study"; (c) HiL control for different test benches: Thun (1984): "Verfahren zum Simulieren von Trägheitsmomenten und geregelter Prüfstand zur Durchführung des Verfahrens [Procedure for simulating moments of inertia and a controlled test bench for validation of the procedure]"; Bayati et al. (2018): "A wind tunnel/HIL setup for integrated tests of Floating Offshore Wind Turbines"

Referee remark (6): Page 3 (65) "As a reference, we use the 3-mass oscillator model of a WT drive train, and for HiL controller synthesis the 3-mass oscillator model of a STB drive train parameterized as by Leisten et al. (2019b)." Please mention the main futures of the WT and drivetrain used as reference case in this study.

Author's response to remark (6): As the topic of the paper are the control methods, we had left out most of the details concerning the reference wind turbine and the device under test in the discussion paper. We now decided, that for sake of interpretation of the results and reproducibility, information is added to the text. Changes in the manuscript: Since information on the reference wind turbine and the device under test is not part of the HiL control methods, but is required for the analytical comparison and simulation results we added it to section 3 (l.101,f):" For the comparison of HiL control methods, we use WT and STB models based upon an NEG Mincon WT with a rated power of 2.75 MW at a rated rotational speed of 17.5 rpm (Leisten et al., 2019). The dominant eigenfrequencies and the drive train parameters have been derived from a multi-body simulation (Baseer et al., 2019). The two dominant eigenfrequencies of the WT drive train are the first edgewise blade mode and the first drive train torsional mode, located at 2.5 and 4.6 Hz, respectively. Drive train parameters of the 3-mass oscillators for WT and STB can be found in the appendix (Table A2)." Please note that Table A2 of the manuscript's appendix is attached to this document

Referee remark (7): Fig 6: please discuss the results more in details, specially why the results for each method differs to others.

Author's response to remark (7): The discussion of analytical results has been expanded. Additionally, the discussion has been extended to a comparison of the respective phase responses. Changes in the manuscript: The discussion for results in Fig. 6 has been replaced by the following text: "Analytical results are presented as transfer functions. Fig. 6 depicts a bode plot for the transfer function from input torque to generator speed for the WT and STB drive trains. The inertia difference and eigenfrequencies of the WT and STB drive trains are observable by an offset towards low frequencies and peaks in the magnitude response, respectively. Additionally, the transfer function for the HiL controlled drive train of the STB is shown for IE and MRC methods. The transfer function with IE method achieves good accordance with the transfer function of the WT up to 1 Hz. The accordance is enabled by the virtual rotor that lowers the magnitude response of the STB until it matches the transfer behaviour of the WT drive train. For frequencies above 1 Hz, the transfer functions mismatch as the first WT drive train eigenfrequency dominates. Thus, a reproduction of the eigenfrequencies seems not achievable with IE method. In contrast to the IE method, the MRC method matches the desired behaviour in the magnitude response up to 6 Hz, despite small deviations. Fig. 3 reveals that the small deviations in magnitude response are caused by the implemented speed controller. The reference model of the WT drive train provides the required WT eigenfrequencies, at the same time eliminating the inertia difference. For frequencies above 6 Hz, the bandwidth of the speed controller becomes a limiting factor and transfer functions of the WT drive train and the STB drive train with MRC method drift apart. However, the reproduction of the WT eigenfrequencies seems possible with the MRC method. The phase response of none of the methods exactly matches the desired transfer function for frequencies above 0.5 Hz. The deviation in phase response is uncritical for HiL operation if WT pitch control provides sufficient phase margin."

Referee remark (8): Please consider including a nomenclature listing all abbreviations.

Author's response to remark (8): In total, six abbreviations are used throughout the

paper. All of them are introduced in the abstract and thus, we avoid the usage of a nomenclature.

[Figure]

**Table A2.** 3-mass oscillator drive train parameters for WT and STB with DUT

| Drive train parameter | WT | STB with DUT | [unit] |
|---|---|---|---|
| $I_1$ | 3.42e6 | 8e4 | [kgm$^2$] |
| $I_2$ | 4.06e6 | 3.3e4 | [kgm$^2$] |
| $I_3$ | 1e6 | 1e6 | [kgm$^2$] |
| $c_{12}$ | 5.2e8 | 3.7e8 | [Nm/rad] |
| $d_{12}$ | 1.3e6 | 2e4 | [Nm/(rad/s)] |
| $c_{23}$ | 6.2e8 | 2.3e8 | [Nm/rad] |
| $d_{23}$ | 1.3e6 | 2e5 | [Nm/(rad/s)] |
| $f_{eig,1}$ | 2.5 | 6.5 | [Hz] |
| $f_{eig,2}$ | 4.6 | 23 | [Hz] |

**Fig. 1.**